# Stabilized Time Transfer via a 1000-km Optical Fiber Link Using High-Precision Delay Compensation System

Bo Liu [1,2,3], Xinxing Guo [1,2,3], Weicheng Kong [1,2,3], Tao Liu [1,2,3,*], Ruifang Dong [1,2,3] and Shougang Zhang [1,2,3]

1   National Time Service Center (NTSC), Chinese Academy of Sciences, Xi'an 710600, China; liubo@ntsc.ac.cn (B.L.); guoxinxing@ntsc.ac.cn (X.G.); kongweicheng@ntsc.ac.cn (W.K.); dongruifang@ntsc.ac.cn (R.D.); szhang@ntsc.ac.cn (S.Z.)
2   University of Chinese Academy of Sciences, Beijing 100039, China
3   Key Laboratory of Time and Frequency Standards, Chinese Academy of Sciences, Xi'an 710600, China
*   Correspondence: taoliu@ntsc.ac.cn

**Abstract:** Variations in optical fiber length and refractive index are induced by environmental perturbation, resulting in an additional dynamic propagation delay in fiber-based time synchronization systems, which deteriorate their transfer stability. This disadvantage can be significantly reduced by transmitting the time signal in both directions through fiber and constructing a feedback loop to compensate the propagation delay at the remote end of the link. This paper proposes an analog-digital hybrid proportional integral derivative (PID) control compensation system based on the time-frequency phase-locked loop (TF-PLL). The system is designed to keep the merits of wide servo bandwidth, servo accuracy, and a large dynamic delay compensation range up to 1 s, which is much greater than that reported in previous studies. For proving the validity of this proposed scheme, a self-developed optical fiber time synchronization equipment based on the delay compensation system is applied. The delay compensation system is used on a 1100-km long laboratory optical fiber, and the results show that the time synchronization stability in terms of time deviation (TDEV) is less than 5.92 ps/1 s and 2.56 ps/10,000 s. After successful laboratory evaluation, the proposed system is installed on a real 988.48-km line between the Xi'an Lintong branch of the National Time Service Center (NTSC) and Linfen City, Shanxi Province, realizing the time synchronization of 10 stations along the optical fiber link. The experimental results in the 988.48-km link illustrate that the measured time difference with a peak-to-peak value of 176 ps, the standard deviation of 19.3 ps, and a TDEV of less than 10.49 ps/1 s and 2.31 ps/40,000 s is achieved. The high-precision time delay compensation system proposed in this paper is simple, reliable, and accurate; has a wide range of compensation; and opens up a feasible scheme for providing synchronized time signals to multiple users over the long-distance field optical fiber networks.

**Keywords:** time synchronization; time delay compensation; fiber-optic link

## 1. Introduction

Successful transmission of time or frequency signals to a remote location is important in many physical and scientific applications [1–7]. In order to obtain better network quality and service experience, there is a range of emerging technologies, e.g., CA/COMP/MIMO and other technologies in the fifth generation (5G) mobile networks that will need high-precise time synchronization. Short-distance fiber-based optic distribution of time and frequency signals has often been considered sufficiently stable, and therefore, an uncompensated one-way transfer has been typically used to realize this transmission [8–12]. However, when fiber distances increase to a few kilometers or even hundreds of kilometers, the frequency stability of a reference signal transmitted through a fiber link can deteriorate due to environment perturbations along the fiber, such as mechanical perturbations and temperature variations, resulting in phase fluctuations [13]. Using the delay compensation

operation in a feedback loop to stabilize the propagation delay of a link can mitigate this problem [14]. For signal transmission through a fiber of thousands of kilometers in length, an increment in the delay fluctuation delta can be significant, which has led to the development of delay compensation techniques [15–19]. The delay compensation technology is based on comparing the remote end signal with the local end signal to extract the phase error signal, which is then used as feedback to tunable devices, such as controlled optical delay lines [20–22] or electrical delay lines [14,23].

Studying the controlled optical delay compensation technology, Wang [20] proposed an optically controlled, continuously tunable, dispersionless optical delay element, whose delay can vary as much as 44 ns. In [21], the frequency transfer over an urban 86-km fiber with a resolution of $2 \times 10^{-18}$ at one-day measuring time was achieved using an optical compensator, and the total dynamic range was 6 ns. However, this optical delay compensation system has a complex structure and is difficult to integrate; also, the compensation range is only 10–100 ns, which is not conducive to engineering applications [24,25]. Changing a large optical delay into a compact electronic delay is desirable because, in this way, the size of the entire transceiver located at the local end can be significantly reduced, thus decreasing the solution's cost-effectiveness and simplifying the operation and maintenance.

Thereafter, Sliwczynski et al. [14] presented an implementable method for stabilizing the propagation delay in a fiber-optic link using two matched electronic delay lines. The measured intrinsic short-term jitter was about 10 ps root mean square (RMS), and the delay was approximately 160 ns. In [23], three different methods for extending the delay compensation range of an optical fiber system with stabilization of propagation delay were presented, and the delay compensation range was increased to approximately 1200 ns. However, an actual delay value of a delay line unit shows certain deviations, which leads to delay uncertainty; so, delay line calibration is necessary [26]. The actual delay value of a delay line unit can also vary with changes in temperature and voltage. Moreover, the problem of a narrow delay control range has to be addressed.

The National Time Service Center (NTSC) plans to establish a high-accuracy ground-based time service system (HAGTS) with a length of 20,000 km to transmit the UTC time signals in China. Therefore, annual delay fluctuations in long-haul links may be substantially higher than short-term delay fluctuations, reaching the order of milliseconds. The compensation range of the optical control delay compensation technology and electronic delay line compensation technology mentioned in the related literature is of the ns-level, which is slightly insufficient. This paper proposes an analog-digital hybrid proportional integral derivative (PID) control compensation scheme based on the time-frequency phase-locked loop (TF-PLL) to transmit a 1-pulse per second (PPS) signal and the time code of coordinated universal time (UTC) at the same time, which can achieve time synchronization for long transmission distances. The compensation range of the proposed delay compensation scheme is larger than 1 s, which overcomes the disadvantage of insufficient compensation range of an electronic delay line and reduces equipment size, energy consumption, and associated costs compared to the cumbersome optical delay compensation. The high-precision time delay compensation system proposed in this paper has been successfully used in optical fiber time synchronization and has achieved good results in laboratory and field link optical fiber testing.

The rest of this paper is organized as follows. Section 2 introduces the block diagram of ultra-long-haul fiber-optic time synchronization and high-precision delay compensation. Section 3 verifies the performance of the delay compensation system by laboratory test on a 1100-km optical fiber link using a homemade 3U prototype. Section 4 analyzes the test results of the 988.48-km field optical fiber link. Finally, Section 5 concludes the paper.

## 2. Operation Principle

### 2.1. Bidirectional Time Transfer

The timing sequence diagram based on the dense wavelength division multiplexing (DWDM) of bidirectional optical fiber time transfer is shown in Figure 1. The presented

scheme uses a bidirectional time comparison method to compensate for the time delay at the remote end dynamically, where the time user is located based on the time delay difference between the local and transmitted signals [14].

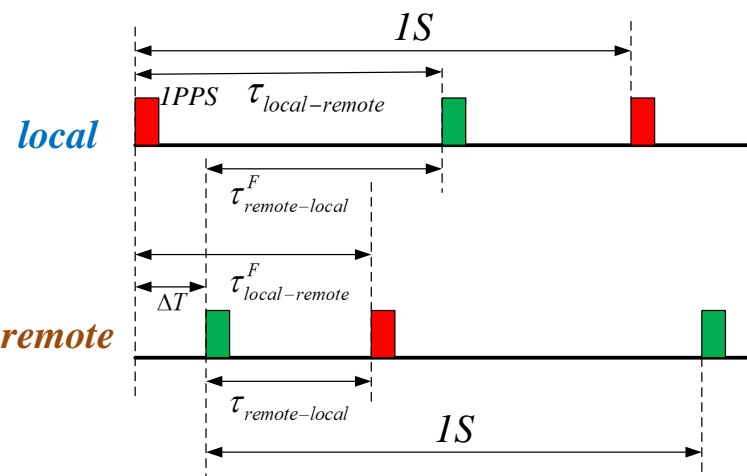

**Figure 1.** Timing sequence diagram based on the dense wavelength division multiplexing of bidirectional optical fiber time transfer.

A time interval counter (TIC) is used to compare the 1-PPS signal from the local end and the 1-PPS signal generated by an oscillator in the remote end [27,28]. Thus, it can be written that

$$\tau_{local-remote} = \Delta T + \tau_{remote}^T + \tau_{remote-local}^F + \tau_{local}^R \tag{1}$$

$$\tau_{remote-local} = -\Delta T + \tau_{local}^T + \tau_{local-remote}^F + \tau_{remote}^R \tag{2}$$

where $\tau_{local-remote}$ and $\tau_{remote-local}$ are the time intervals measured between the local end and the remote end and vice versa, respectively; $\Delta T$ is the instantaneous clock difference between the local and remote ends; $\tau_{local-remote}^F$ and $\tau_{remote-local}^F$ are the forward and backward propagation delays of a fiber connecting the local and remote ends; $\tau_{local}^R$ and $\tau_{remote}^R$ are the receiving equipment delays of the local and remote ends, respectively; and $\tau_{local}^T$ and $\tau_{remote}^T$ denote the sending equipment delays of the local and remote ends, respectively.

The value of $\Delta T$ can be calculated as follows:

$$\begin{aligned} 2\Delta T = \tau_{local-remote} - \tau_{remote-local} + \tau_{local-remote}^F \\ - \tau_{remote-local}^F + \tau_{local}^T - \tau_{local}^R + \tau_{remote}^R - \tau_{remote}^T \end{aligned} \tag{3}$$

Using equal wavelengths of the lasers in bidirectional transmission can eliminate the delay mismatch caused by the chromatic dispersion and temperature-induced variations. This idea relies on the assumption that the propagation delays in the two directions are equal and equally depend on environmental conditions, which makes the second righthand term in Equation (3) equal to zero ($\tau_{local-remote}^F = \tau_{remote-local}^F$). Thus, ideally, the clock comparison is not affected by the fiber propagation delay. Based on the time interval data measured at the local and remote ends, the instantaneous clock difference at both ends can be obtained, and then, the clock synchronization at the two ends can be achieved.

However, in the equal-wavelength scheme, the desired far-end generated signal cannot be separated from the Rayleigh backscattering signal originating from a local transmitter. This is the dominant performance-limiting factor in bidirectional, single-wavelength systems [29]. To avoid the noise resulting from the superposition of the backscattered and main signals at receivers, lasers are detuned at approaching 0.8 nm (100 GHz). In this way, all noise resulting from suppressing the received signal by the backscattered light can be filtered from the receiver bandwidth. Laser detuning and

variations in environmental temperature introduce certain differences in the propagation delay for signals traveling through fiber in opposite directions ($\tau^F_{local-remote} \neq \tau^F_{remote-local}$).

For a standard single-mode fiber (SMF) link with a length of L, the propagation delay can be expressed as follows:

$$\tau^F_{local-remote}, \tau^F_{remote-local} = \frac{L}{c}\left(n - \lambda\frac{dn}{d\lambda}\right) \tag{4}$$

where $n$ is the fiber refractive index, $c$ is the speed of light in vacuum, and $\lambda$ is the transmitting wavelength [30].

The refractive index $n$ can be empirically fitted by the Sellmeier equation as follows:

$$n^2 = A + \frac{B}{1 - C/\lambda^2} + \frac{D}{1 - E/\lambda^2} \tag{5}$$

where the Sellmeier coefficients $A$, $B$, $C$, $D$ and $E$ have been empirically fitted with respect to temperature, $T$, for different glasses [4]. According to Equation (5), the delay of optical signals with different wavelengths transmitted through an optical fiber link with the same length is inconsistent. It should be noted that transmission delay is a function of temperature.

Next, deriving Equation (4) with respect to the temperature, where $\left.\frac{d\left(\tau^F_{local-remote}, \tau^F_{remote-local}\right)}{dT}\right|_\lambda$ may be rewritten as follows:

$$\left.\frac{d\left(\tau^F_{local-remote}, \tau^F_{remote-local}\right)}{dT}\right|_\lambda = \frac{1}{c}\left[\frac{dL}{dT}\left(n - \lambda\frac{dn}{d\lambda}\right) + L\left(\frac{dn}{dT} - \lambda\frac{d^2n}{d\lambda dT}\right)\right] \tag{6}$$

Therefore, the relationship between the time delay difference and temperature of optical signals transmitted at two wavelengths through an optical fiber link is as follows:

$$\begin{aligned}
\left.\frac{d\left(\tau^F_{local-remote}, \tau^F_{remote-local}\right)}{dT}\right|_{\lambda^{local}-\lambda^{remote}} &= \frac{1}{c}\frac{dL}{dT}\left(n_{\lambda^{local}} - n_{\lambda^{remote}}\right) \\
&+ \frac{L}{c}\frac{d}{dT}\left(\left(n_{\lambda^{local}} - n_{\lambda^{remote}}\right) - \lambda^{local}\frac{dn_{\lambda^{local}}}{d\lambda^{local}} + \lambda^{remote}\frac{dn_{\lambda^{remote}}}{d\lambda^{remote}}\right) \\
&+ \frac{1}{c}\frac{dL}{dT}\left(\lambda^{remote}\frac{dn_{\lambda^{remote}}}{d\lambda^{remote}} - \lambda^{local}\frac{dn_{\lambda^{local}}}{d\lambda^{local}}\right)
\end{aligned} \tag{7}$$

Considering the common thermal coefficient of the propagation delay of below 40 ps/km*K [15] and assuming seasonal temperature variations of approximately 45 K, a delay tuning of 1.8 ns per kilometer is required for a link [16]. Thus, variations in the fiber temperature can cause the dominant accuracy limit in time synchronization.

## 2.2. High-Precision Delay Compensation

The proposed analog-digital hybrid PID control scheme based on the TF-PLL compensation includes mainly two parts. The first part is a TF-PLL system. In time transfer, a 1-PPS signal might trigger the insertion of a certain pattern (e.g., a pseudo-noise sequence) or introduce a phase variation into the periodic pulse train. As shown in Figure 2a, in the block diagram of the TF-PLL system, a phase detector (PD) calculates an error signal $p$ between 10-MHz signals $f_1$ and $f_2$ output by the VCO and clock, respectively. The 10-MHz frequency signal output by a high stability crystal oscillator $f_2$ is input to a frequency divider, and a 1-PPS time signal $t_2$ is generated by the frequency divider. At the same time, the remote end of time transmission outputs a 1-PPS time signal $t_1$ deteriorated due to the accumulated noise in a long-distance link. Then, time signals $t_2$ and $t_1$ are input to a time interval measurement module at the same time to obtain the time difference $\varepsilon$ between them. The proportional integral modulator (PIM) controls the high-temperature crystal oscillator to adjust to the phase difference signal $p$ and the time difference analog signal

until the output frequency signal $f_2$ and time signal $t_2$ are highly stable and synchronized with signals $f_1$ and $t_1$ output by the time-frequency source. A loop filter (LF) is used to attenuate a high-frequency component in the error signal.

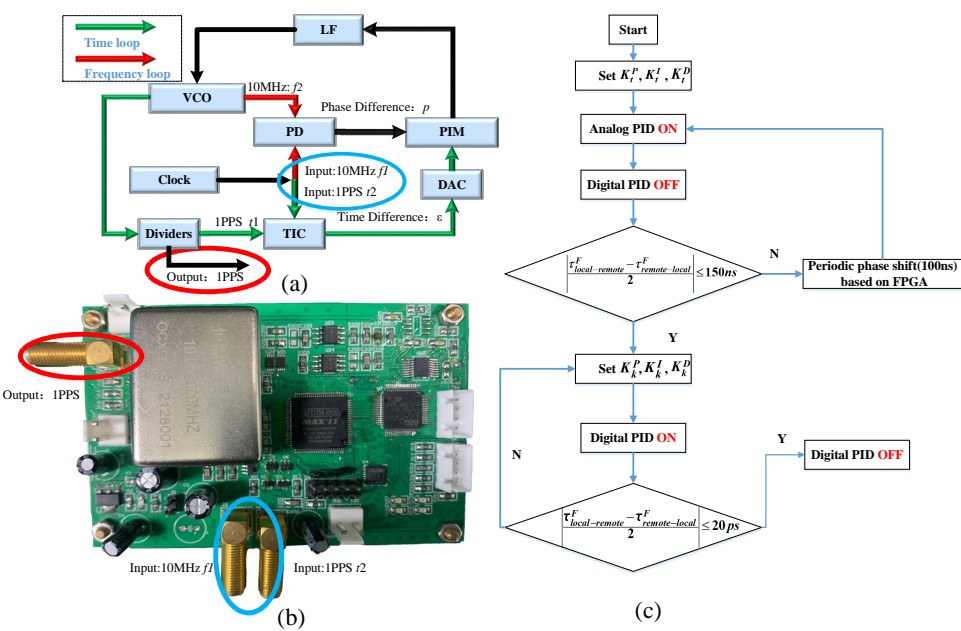

**Figure 2.** (**a**) Block diagram of the TF-PLL system; (**b**) Delay compensation system on a PC board. (**c**) Flowchart of the analog-digital hybrid PID control system.

The continuous-frequency PLL is added to the discrete-time PLL to suppress the interference of the 1-PPS pseudo-noise sequence. The entire system forms a time-frequency bilayer delay-locked loop, which forces the phase difference between the input and the feedback frequency signal to zero, regardless of fiber delay fluctuations. Using the measured data of the time loop as a feedback input to the frequency loop, even if the time-delay drift of temperature is introduced into the compensation system, the drift error can be obtained and corrected through precise time interval measurement, and the time-delay drift can be suppressed.

For the second part, as shown in Figure 2c, when analog PID is used to control the frequency PLL, the analog error signal collected by the PD is used to control the VCO through the LF. The output of the analog PID control system is given by

$$u_t^{out} = K_t^P e(t) + K_t^I \int e(t)dt + K_t^D \frac{de(t)}{dt} \tag{8}$$

where $u_t^{out}$ is the output at time t; $e(t)$ is an analog error signal at time t; $K_t^P$ is the analog proportional term, which makes the output signal gradually approach the target signal; $K_t^I$ is the analog integral term, which is used to eliminate the fixed error between the output value and the target value after the system becomes stable; and $K_t^D$ is the analog differential term, which plays a role in adjusting the change rate of the error value and in changing the adjustment speed of the whole control system.

Therefore, to obtain a faster response time and to eliminate static errors as much as possible, the values of $K_t^P$ and $K_t^I$ should be slightly larger than the normal parameter settings, and the $K_t^D$ value should tend to zero when setting analog PID parameters.

In the time PLL, a 1-PPS phase difference greater than 100 ns needs to be compensated by a fully periodic phase shift using a field-programmable gate array (FPGA). The periodic phase shift realized by an FPGA can provide a wide-range time delay control; the control range is larger than 1 s. However, if a 10-MHz signal is used as the reference clock of an FPGA, the phase can be shifted only by an integer multiple of 100 ns. For a 1-PPS phase

difference of less than 100 ns, the digital PID is used to control the time PLL. The error signal is digitized, processed by a digital PID algorithm, and then converted back to the analog signal by a digital-to-analog converter (DAC) to control the VCO. The output of a PID control system is expressed as follows:

$$u_k^{out} = K_k^P e_k + K_k^I \sum_{j=0}^{K} e_j + K_k^D \frac{e_k - e_{k-1}}{T} \tag{9}$$

To improve the control speed and reduce the operation time, it should hold that

$$u_k^{out} = u_k^{out} + \Delta u_k^{out} \tag{10}$$

where $\Delta u_k^{out}$ denotes an increment from $u_{k-1}^{out}$ to $u_k^{out}$, and $\Delta u_k^{out}$ in a digital PID controller is realized by a high-precision DAC.

In a certain range, there is a linear relationship between the oscillation frequency of a VCXO and the control voltage, and the accuracy of a DAC directly affects the control accuracy of the oscillation frequency.

For the frequency control of (10 MHz ± 2.5 Hz) of a VCXO, the variation range of control voltage is 0–3.3 V. To meet the design requirements, i.e., to achieve an output frequency accuracy of better than, the resolution of control voltage should satisfy the following condition:

$$R_v = 10^{-11} \times 10^7 \times \frac{5}{2.5 \times 2} = 1 \times 10^{-4} \text{ V} \tag{11}$$

The resolution of the control voltage is determined by the output voltage range and bits of the DAC. In fact, when the output voltage is in the range of 0–5 V, and the number of DAC bits is n, the resolution can be expressed as follows:

$$R_{DAC} = 2 \times \frac{5}{2^n} = 5 \times 2^{1-n} \tag{12}$$

Comparing Equations (11) and (12), it can be obtained that $n \geq 14$. In digital PID, it takes a certain time to adjust a VCXO using a DAC to stabilize the output frequency with a large lag. Moreover, the adjustment range of a frequency should not be large, and the adjustment cycle should be long, that is, $K_k^P$ and $K_k^I$ values should not be too large, while $K_k^D$ value should be large.

Therefore, the total output of the hybrid analog-digital PID control based on the TF-PLL compensation is given by

$$u^{out} = c_1 u_t^{out} + c_2 u_k^{out}$$
$$s.t. \begin{cases} c_1 + c_2 = 1 \\ 0 \leq c_1 \leq 1 \\ 0 \leq c_2 \leq 1 \end{cases} \tag{13}$$

where $c_1$ and $c_2$ are analog-to-digital distribution scale coefficients, and in this paper, they are both set to 0.5.

## 3. Experimental Evaluation

### 3.1. Delay Compensation Uncertainty and Stability Analysis

In the first experiment, the fundamental feature of the proposed system, which is the uncertainty of the fiber propagation delay compensation control, was demonstrated.

The experimental device used to verify the proposed delay compensation method is shown in Figure 3a. The AFG31052 produced by Tektronix was used as a signal source to generate a 10-MHz frequency signal and a 1-PPS time signal, which were then input to the SR620 produced by Stanford as the reference frequency and stop signals. The signal source output another homologous 1-PPS signal, which was input to the delay compensation

control module. After the control system set the delay n, a 1-PPS signal was generated as a start signal of the SR620.

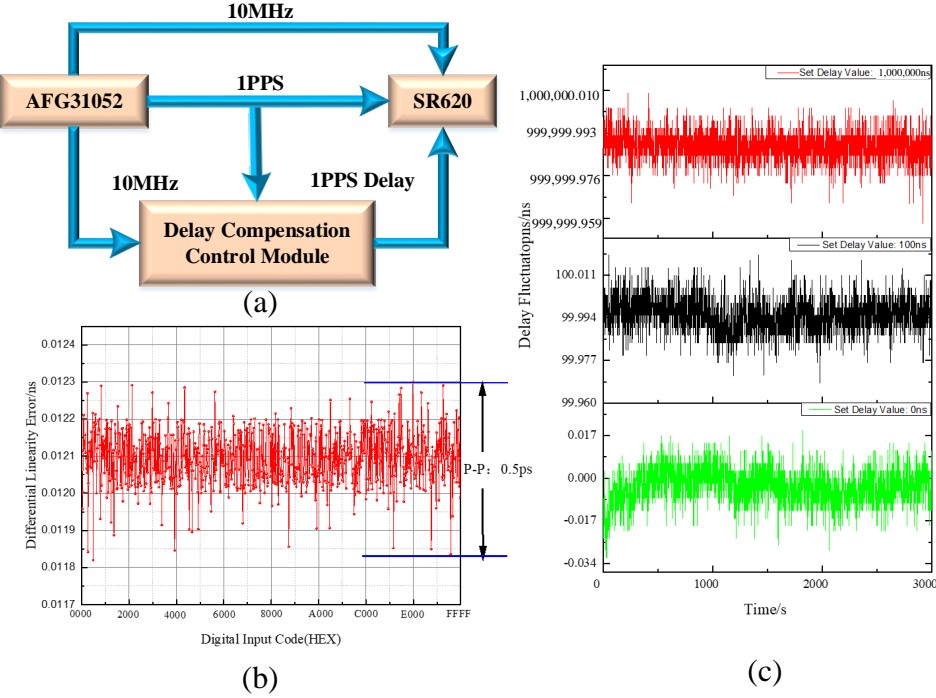

(a)

(b)

(c)

**Figure 3.** (**a**) Test diagram of the delay compensation control module; (**b**) Differential linearity error vs. code; (**c**) Delay fluctuations of the delay compensation control system.

In the development of the delay compensation hardware system, a TDC-GP21 produced by ACAM was used as a time interval measurement chip, and the RMS was 50 ps. In addition, a 16-bit DAC8531, which is a low-power, single, 16-bit buffered voltage output DAC, was used. The changing trend of the differential linearity error of the compensation system with the DAC digital input code is shown in Figure 3b; the corresponding phase modulation range was 800 ns.

The system set the delay compensation control value x from 0 to 10 s and collected the test data of the SR620 for 1 h. The average value of the collected data was taken as the measured delay compensation control value. The records showing the delay fluctuations of the delay compensation control system are shown in Figure 3c.

As shown in Table 1, when the delay compensation was set to zero, the measured time interval was 23.576 ns. This deviation was mainly introduced by the coaxial line delay, the chip transmission delay, and the SR620 measurement system. The time interval data were generally uniform without obvious fluctuation. After deducting the influence of the system deviation, it was calculated that the time deviation of the time interval under each time delay setting value was within ±25 ps, which was basically equivalent to the deviation introduced by the nonlinearity of the SR620 measurement. Theoretically, the time delay control range is infinite. When the time delay value was set to 0.1 s, the time deviation could be controlled within 25 ps.

We have conducted experiments to verify the effect of temperature on the performance of the time delay compensation system. As shown by the red curve in Figure 4a, the whole test system was in the program-controlled temperature box; the temperature change was from 15 °C to 30 °C between two cycles within 1 h, and in the measurement period, variations of more than 15 °C in air temperature were observed. In Figure 4a, the black curve shows the variations in the propagation delay of the fiber over a 400-ps period. To solve the influence of temperature on the system, temperature control was realized for the delay compensation system. The blue curve in Figure 4a shows that the delay

compensation control system could maintain excellent stability under an air temperature change of 15 °C.

**Table 1.** Delay compensation control uncertainty results.

| Set Delay Value (ns) | Measured Delay Value (ns) | Time Deviation After Calibration (ns) |
|---|---|---|
| 0 | 23.576 | 0 |
| 1 | 24.592 | +0.016 |
| 100 | 123.595 | +0.019 |
| 10,000 | 10,023.553 | −0.023 |
| 1,000,000 | 1,000,023.601 | +0.025 |
| 100,000,000 | 100,000,023.555 | −0.021 |

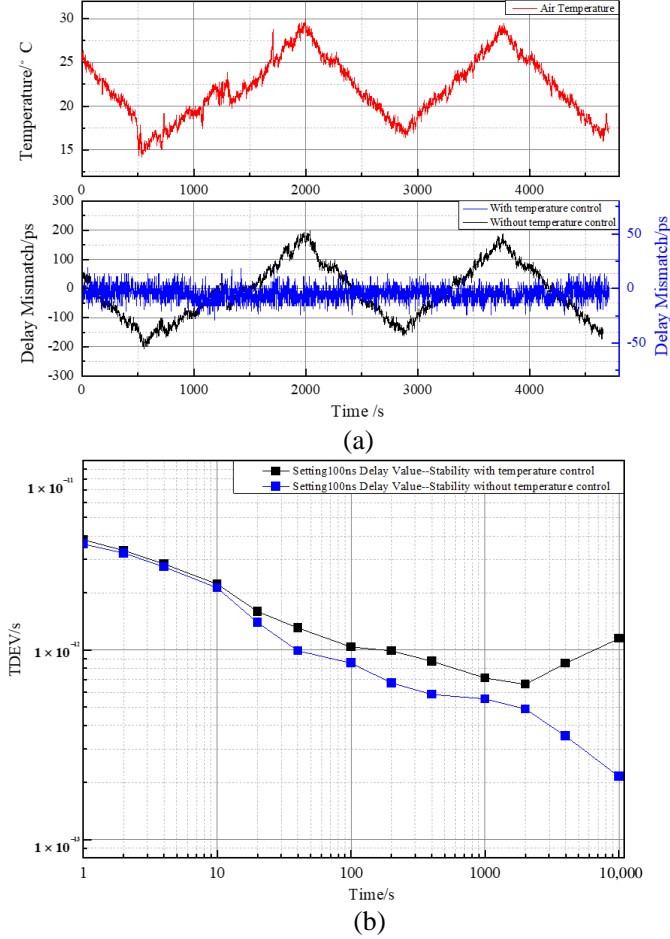

**Figure 4.** (**a**) Impact of temperature on the time delay compensation system performance. (**b**) TDEV results of the delay compensation control system for a delay of 100 ns.

Further, to analyze the stability of the compensation system, the delay was set to 100 ns, and 24-h data were collected. The stability of the 1-PPS signal output when the processed delay compensation control module was applied is shown in Figure 4b. The black curve in Figure 4b shows the TDEV results of the uncontrolled temperature delay compensation system, in which the TDEV was 4.1 ps/1 s and 1.14 ps/10,000 s; the blue curve shows the TDEV results of the temperature control delay compensation system, which showed a TDEV of 3.8 ps/1 s and 0.36 ps/10,000 s. Therefore, the benefit of the temperature control of the delay compensation system for long-term stability was obvious.

### 3.2. Bidirectional Time Transmission Analysis

After verifying the delay compensation system, the proposed system was applied to optical fiber time synchronization, and its performance was tested. The measurements were performed using the setup shown in Figure 5a, and both local and remote parts of the system were located in the same laboratory. A 1100-km optical fiber time synchronization system link was constructed in the laboratory. The optical fiber used in the experiment was a 50-km bundle of G.652 spooled fibers, with a total of 22 bundles and a 100-km optical fiber between each station.

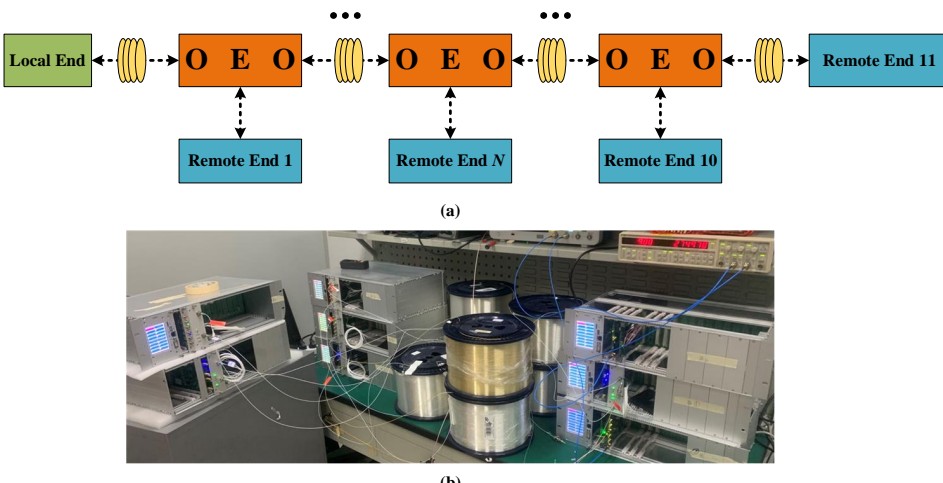

(a)

(b)

**Figure 5.** (**a**) Configuration of the fiber-optic link used in the experiment. (**b**) The experimental bench schematic.

In the development of time synchronization equipment, it was difficult to ensure the consistency of the performance parameters of each piece of equipment. The asymmetry of the optical path in equipment and different transmission delays of the circuit could lead to different degrees of delay for each piece of equipment. As show in Figure 5b, the 1100-km laboratory optical fiber link was divided into 10 segments. To ensure the time synchronization transmission accuracy between stations, the parameters of the equipment were adjusted at each station, and the time delay difference was calibrated.

The system used 11 SFP lasers operating near 1548.51 nm (C36) and 21 SFP lasers operating near 1549.32 nm (C35). Since the chromatic dispersion accumulated along the optical fiber influenced both the propagation delay and the shape of the received optical signal, the laser transmitters were equipped with wavelength lockers and electro-optical external modulators.

As shown in Figure 6a (C35 channls) and Figure 6b (C36 channls), the wavelengths of all 32 lasers were measured and calibrated one-by-one using a wavelength meter, and the error was less than 0.0003 nm. To avoid the influence of laboratory temperature change (peak-to-peak value of 2 °C/day [31]) on the output wavelength of a laser, the external temperature of the laser was controlled to ensure the change in the output wavelength with ambient temperature was less than 0.0001 nm/°C.

Figure 7a shows the measured results of the 0–1100 km optical fiber time synchronization. For comparison, the fiber-optic time synchronization measurement was performed by replacing each fiber span with a 1-m fiber. This measurement result was regarded as the noise floor (zero) of the experimental apparatus. The red curve in Figure 7a shows that when the system was on the noise floor of the optical fiber time synchronization link, the peak-to-peak value was 56 ps, and the standard deviation was 9.35 ps; the black curve shows that in the case of a 1100-km optical fiber, the peak-to-peak value of the system was 108.34 ps, and the standard deviation was 15.45 ps, indicating that time synchronization was successfully achieved.

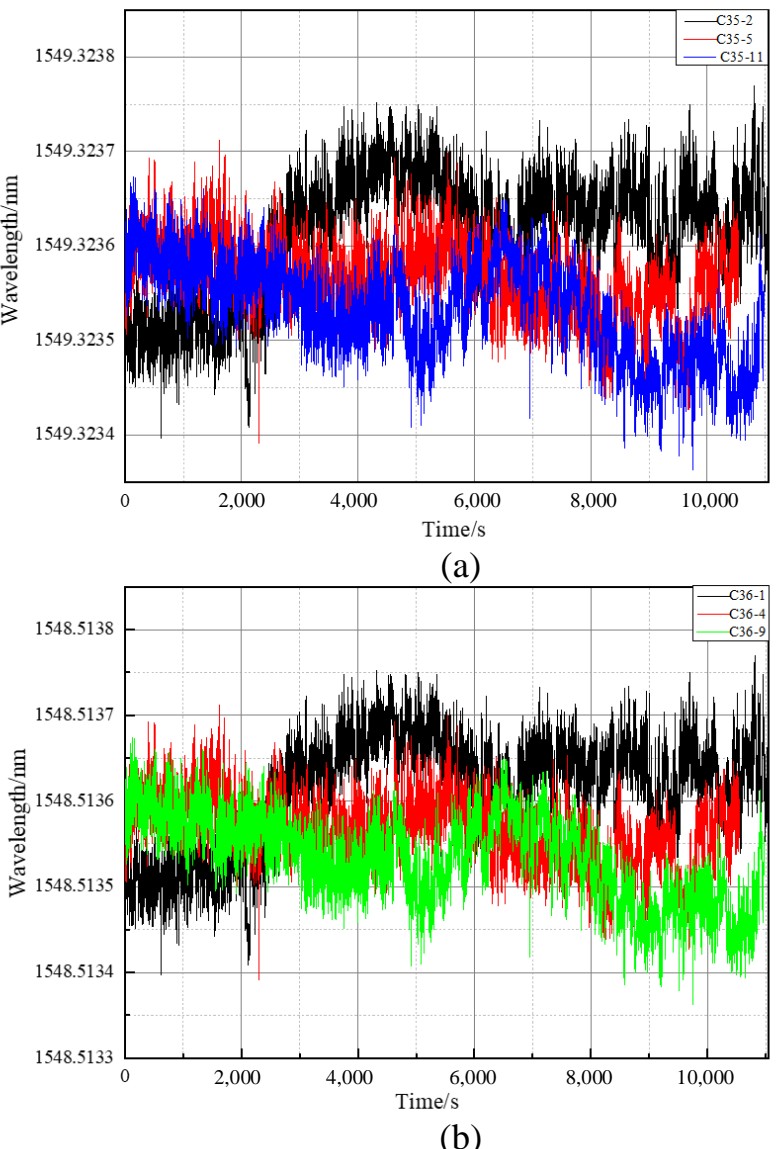

**Figure 6.** Measured wavelength variations of the SFP transceiver for C35 and C36 channels.

To examine the system performance as a function of the main fiber link length, the TDEV was measured at different fiber link lengths. As shown in Figure 7b, the stability of the electrical noise floor (zero) of the measurement was 4.15 ps/1 s and 0.586 ps/10,000 s. Then, the stability deteriorated as the noise accumulated along the fiber link, and the stability at 1100 km was 5.92 ps/1 s and 2.56 ps/10,000 s. The bump in the graph near t = 100 s was due to interference with the air-conditioning system installed in the laboratory and working with a 5-min cycle. The residual bump might suggest the influence of temperature on the measurement equipment despite a relatively stable temperature in the laboratory. When the laboratory environment temperature changed, the delay of equipment and laser wavelength of each station also changed, which caused a certain degree of deterioration in the long-term stability.

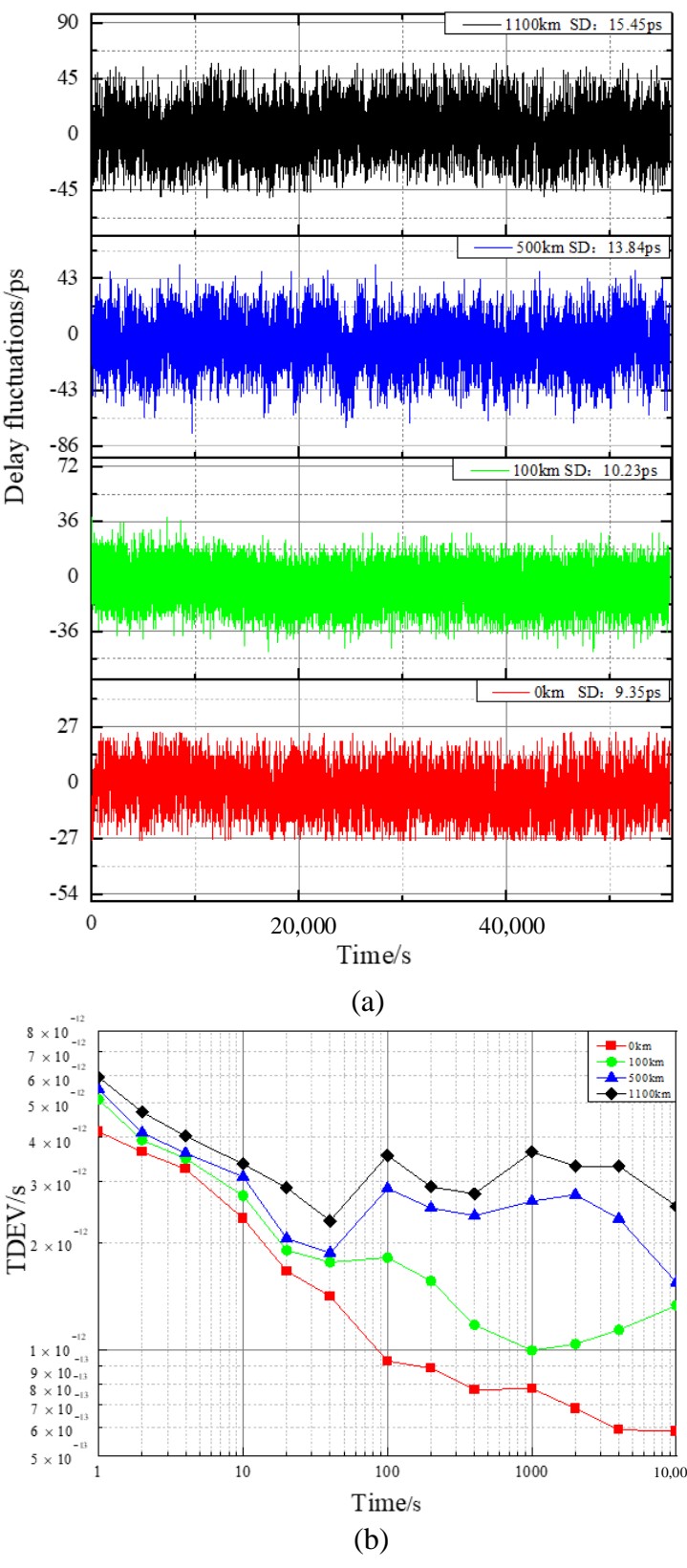

(a)

(b)

**Figure 7.** (**a**) Instability of the propagation delay of the optical fiber link with a length of 0–1100 km. (**b**) TDEV results of the 1-PPS time transfer for different length laboratory-tested links with the propagation delay stabilization.

## 4. Field Test Results

In the next experiment, the proposed system was tested on a real fiber network, which is part of the standard telecommunication infrastructure in the urban environment. As shown in Figure 8, a fiber loop provided by China Telecom, running from the laboratory at the Lin Tong Campus of NTSC, leading to the Lin Tong Telecom engine room, Er Chang, Yan Liang, Pu Cheng, He Yang, Han Cheng, He Jin, Xin Jiang, Lin Fen, and returning back to the laboratory, was used.

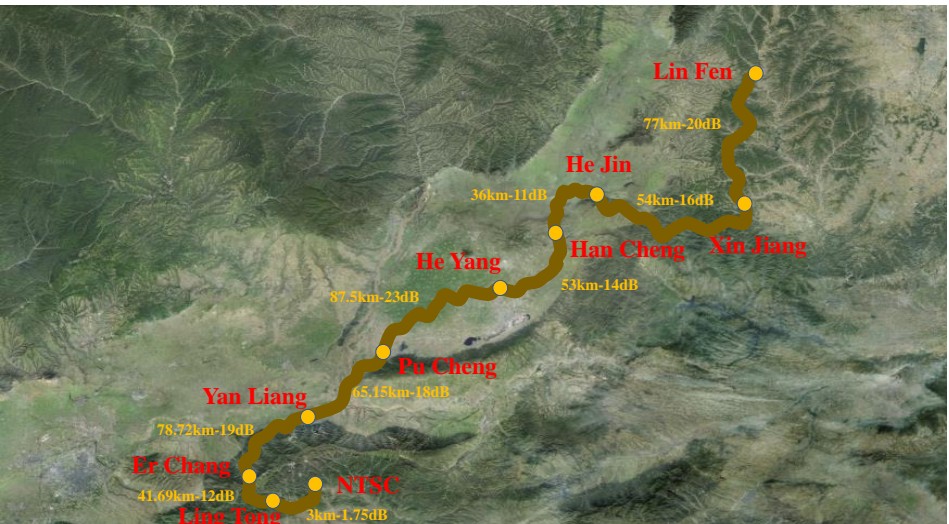

**Figure 8.** Route of the 988.52-km field optical fiber link.

To verify the reliability of the self-developed time synchronization equipment in practical application, the ps-time synchronization transmission was realized using a field optical fiber link with a length of 1000 km. The actual geographical location distribution of the link is shown in Figure 8. The optical fiber type of the field link was G.652, which was consistent with that used in the laboratory test. The 988.52-km-long loops were composed of 10 stations connected by SC-APC connectors, with a relatively large total attenuation of 269.5 dB. The fiber cable was in part placed in underground telecommunication ducts and in part buried along the city bypass highway. Thus, the cable was well-shielded from short-term fluctuations in the external temperature.

The instability of the propagation delay of the 988.52-km field optical fiber link is shown in Figure 9a. The experimental results show that the measured time difference varied with the peak-to-peak value of 176 ps near the SD of 19.3 ps.

The TDEV results of the 1-PPS time transfer via the 988.52-km field optical fiber link are shown in Figure 9b, where it can be seen that the time stability of less than 10.49 ps/1 s and 2.31 ps/40,000 s was achieved. In addition, the measurement results show that the standard difference and time stability of the time synchronization results of the field optical fiber link deteriorated significantly from the measurement results of the laboratory optical fiber link. This was because the field optical fiber link was strongly affected by mechanical vibrations, sound, and various noises, and the spectrum of the noise was wide. Thus, it was difficult to eliminate the noise completely by using the photoelectric optical relay and DLL purification, which had a certain impact on the time synchronization result. Further, the measurement results indicate that the time stability of the field optical fiber link in the first 100 s was significantly poorer than that of the laboratory optical fiber link, i.e., there was short-term stability of time synchronization. Moreover, due to the change in the measurement environment temperature, the output wavelength of the laser and the transmission delay of the electronic circuit in the local end, the remote end, and the relay equipment also changed, which had a severe impact on the long-term stability of time

synchronization. According to the results of the time stability measurement, the stability severely deteriorated after 1000 s.

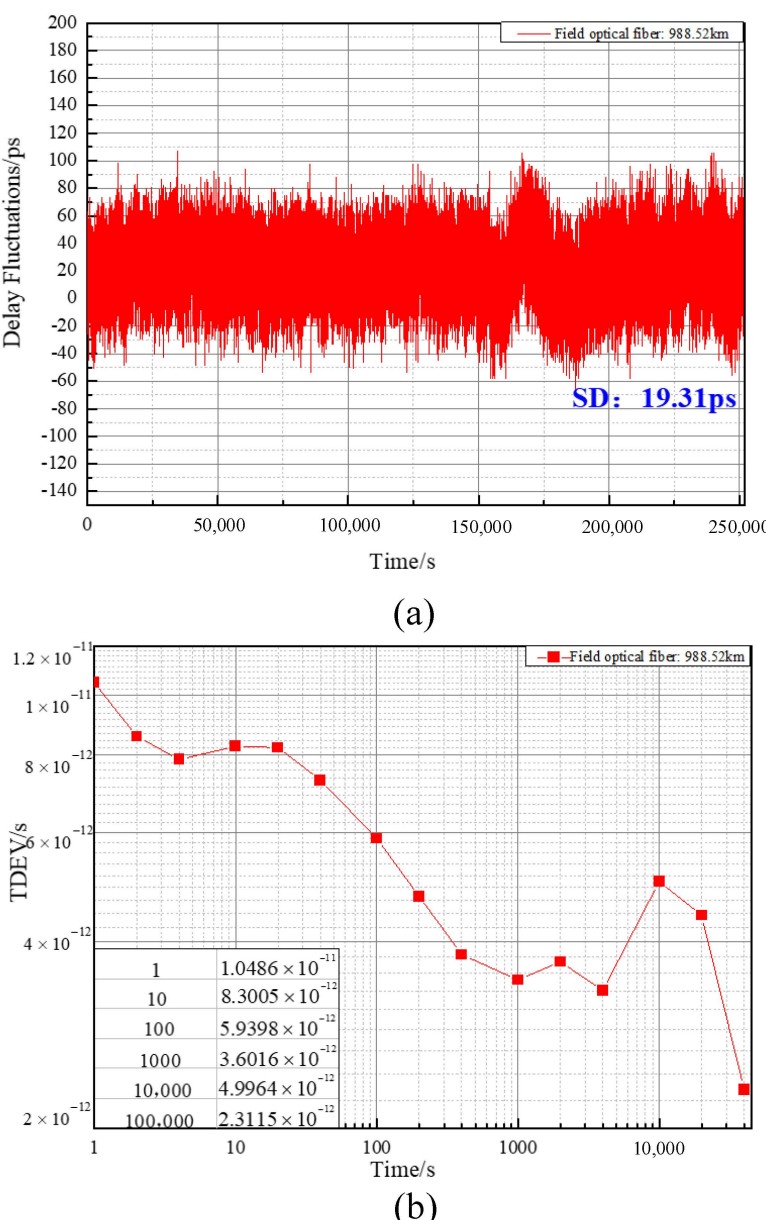

**Figure 9.** (**a**) Instability of the propagation delay of the 988.52-km field optical fiber link. (**b**) TDEV results of the 1-PPS time transfer via the 988.52-km field optical fiber link.

## 5. Conclusions

In conclusion, we propose a hybrid analog-digital PID control compensation system based on the TF-PLL for realizing highly stable time signal synchronization. The high-precision time delay compensation technology proposed in this paper can be applied to the optical fiber time synchronization system, and it can provide China's highest precision time signals for basic scientific research fields, commercial projects, and many engineering applications.

First, the performance of the delay compensation system is tested and analyzed. The compensation range of the system is more than 1 s, and its time interval deviation is within ±25 ps. However, due to the great influence of temperature on the time delay compensation system, we have achieved good results by controlling the temperature on the hardware equipment. Thus, the delay compensation system is used in optical fiber time

synchronization to perform two parts of testing. In the first part, the time synchronization TDEV measured in the 1100-km optical fiber in the laboratory is less than 5.92 ps/1 s and 2.56 ps/10,000 s. In the second part, the time synchronization stability TDEV is less than 10.49 ps/1 s and 2.31 ps/40,000 s in the 988.82-km field optical fiber link from NTSC to Linfen City, Shanxi Province.

However, there are some problems in the high-precision delay compensation system. The resolution of time delay control is determined by the number of DAC bits. In the future, we will try to adopt higher digit DAC to improve the resolution of delay control. The linearity of time delay control is directly related to the linearity of phase detector, which puts forward higher requirements for phase detector. The speed of time delay control is related to the loop control parameters of phase-locked loop. If the speed of time delay control exceeds the speed of loop control, it is easy to lose lock temporarily, which introduces jitter and even whole period error.

The proposed high-precision delay compensation system with improved temperature control will play an important role in the HAGTS project, which aims to establish a length of 20,000 km field fiber network for transmitting the UTC time signals in China.

**Author Contributions:** Conceptualization, B.L. and T.L.; methodology, B.L. and T.L.; formal analysis, X.G.; investigation, W.K.; supervision, T.L., R.D. and S.Z. All authors have read and agreed to the published version of the manuscript.

**Funding:** This work was supported by the National Key Research and Development Program of China (2016YF-F0200200); National Natural Science Foundation of China (NSFC) (11803041, 61127901, 91636101); Strategic Priority Research Program of the Chinese Academy of Sciences (CAS) (XDB21000000); Open Research Fund of State Key Laboratory of Transient Optics and Photonics (SKLST201909).

**Institutional Review Board Statement:** Not applicable.

**Informed Consent Statement:** Informed consent was obtained from all subjects involved in the study.

**Data Availability Statement:** The C programs and case analysis data used to support the findings of this study are available from the corresponding author upon request.

**Conflicts of Interest:** The authors declare no conflict of interest.

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
