# Peer review of "Stabilized Time Transfer via a 1000-km Optical Fiber Link Using High-Precision Delay Compensation System"

_photonics, doi:10.3390/photonics9080522_

Round 1
Reviewer 1 Report
1. Literature Review needs to be extended in the introduction. It is minimal.
2. Some background of this work in accordance with 5G should be given.
3. The theoretical formulation of the method is minimal and not discussed.
4. The setup needs to be explained with a hierarchy or pseudo code. Also, add a table of all the used components and their parameters.
5. The critical discussion on the results and future work is missing.
6. Results in terms of error vector magnitude should be included for:
a. Changing RF input power
b. Received optical power
c. Freq fluctuation vs time
d. correlation of transmitted and received signal
e. It is critical to compare compensated and non compensated signals received after 1000km.
7. The proposed system should be compared with other competitive methods.
8. Why PID method was used? what was the motivation for this?
9. What sort of signal was used? Is it 5G NR? PRS, PSS/SSS (LTE) or CSIRS? What is its bandwidth?
10. Is the synchronization procedure the first path of arrival based or maximum strength time of arrival?
11. THe experimental bench schematic is missing. It should be added.
Author Response
We appreciate you very much for the positive and constructive comments and suggestions

Reviewer 2 Report
The paper proposes a delay compensation scheme based on a time-frequency phase-locked-loop to simultaneously transmit an 1-pulse-per-second signal and the time code of coordinated universal time to achieve time synchronization for long transmission distances. The proposed scheme is verified by testing it, first on a lab fiber link (about 1100km long) and then on an actual fiber link about 1000km long.
The paper is generally well written and has the merit of presenting results from an experimental and an actual link.
Sellmeier equation (5) actually refers to n2 (the refractive index squared) and, for glass, it usually includes three (3) fractional terms. In any case, a short statement should be added regarding what the coefficients A, …, E of eq. (5) represent.
Regarding experimental evaluation (sections 3 and 4) concise tables containing the basic facts of each experiment would be helpful.
I suggest the “Conclusion” section to be rephrased regarding the following:
- Lines 306-310 are rather obsolete.
- The 2nd paragraph (lines 316-315) should be enhanced and present the main findings of the work.
- The 3rd paragraph should not be referred to as ”future work” but rather as a potential application of the work presented in the paper.
Minor editing is needed regarding the use of English (e.g. the word “derive” in line 121, “aiming” in line 327).
Author Response
We appreciate you and reviewers very much for the positive and constructive comments and suggestions

Round 2
Reviewer 1 Report
Comments have been addressed. Paper is acceptable
This manuscript is a resubmission of an earlier submission. The following is a list of the peer review reports and author responses from that submission.